# Computational Fluid Dynamics Modeling of the Resistivity and Power Density in Reverse Electrodialysis: A Parametric Study

**DOI:** 10.3390/membranes10090209

**Published:** 2020-08-29

**Authors:** Zohreh Jalili, Odne Stokke Burheim, Kristian Etienne Einarsrud

**Affiliations:** 1Department of Energy and Process Engineering, Norwegian University of Science and Technology (NTNU), 7491 Trondheim, Norway; zohreh.jalili@ntnu.no; 2Department of Materials Science and Engineering, Norwegian University of Science and Technology (NTNU), 7491 Trondheim, Norway; kristian.e.einarsrud@ntnu.no

**Keywords:** reverse electrodialysis, computational fluid dynamics, power density, factorial design

## Abstract

Electrodialysis (ED) and reverse electrodialysis (RED) are enabling technologies which can facilitate renewable energy generation, dynamic energy storage, and hydrogen production from low-grade waste heat. This paper presents a computational fluid dynamics (CFD) study for maximizing the net produced power density of RED by coupling the Navier–Stokes and Nernst–Planck equations, using the OpenFOAM software. The relative influences of several parameters, such as flow velocities, membrane topology (i.e., flat or spacer-filled channels with different surface corrugation geometries), and temperature, on the resistivity, electrical potential, and power density are addressed by applying a factorial design and a parametric study. The results demonstrate that temperature is the most influential parameter on the net produced power density, resulting in a 43% increase in the net peak power density compared to the base case, for cylindrical corrugated channels.

## 1. Introduction

The energy economy is facing its most challenging decade, as it must transcend into a more climate-friendly one, as half of the emitted CO${}_{2}$ due to energy generation and consumption has been targeted for reduction. To achieve this, the technologies used must be changed from those depending on the burning of fossil fuels into electricity and heat, towards technologies which provide electricity and store it in the form of chemical energy. Striving for renewable energy generation, energy storage systems, and renewable hydrogen production, reverse electrodialysis is one of the few technologies that could address all three of these needs [1,2,3].

Salinity gradient energy (SGE)—particularly RED, which harvests energy produced by mixing two aqueous solutions with different salinities,—has received great interest in the literature [2,3,4,5,6,7,8,9,10,11] since its first use, which was reported by Pattle in 1954 [12]. Concentration batteries have also been recently proposed and discussed, which couple salinity gradient energy (SGE) technologies for energy generation to their corresponding desalination technologies [2,4,13]. Jalili et al. developed mathematical models to compare three types of energy storage systems: electrodialytic, osmotic, and capacitive batteries [2]. Influential parameters, such as temperature and energy consumption of the pump, on the performance of different concentration batteries were also discussed in their work [2] applying a mathematical model. They reported that the peak power densities of the energy storage systems increase at elevated temperature [2].

A schematic of a simple RED stack is shown in Figure 1. In general, a unit cell consists of a dilute solution compartment, a concentrated solution compartment, a cation exchange membrane, and an anion exchange membrane. By repeating unit cells and connecting the end points of the stack to an anode and a cathode compartment (where the electrode rinse solutions are present), a RED stack can be completed for converting an ionic flux into an electrical one [2].

The electrical potential of a RED unit cell is always lower than the open-circuit potential, due to the ohmic resistance, concentration changes in the boundary layer, and concentration changes in the bulk solutions. The last two sources can be interpreted as non-ohmic resistances [5,14]. Non-ohmic resistance is mainly controlled by concentration polarization [15], which has been investigated and discussed by several researchers in the literature [15,16,17,18,19,20,21].

Although it has been agreed, by some researchers that increasing the flow velocity and the introduction of flow promoters (i.e., spacers) can mitigate the concentration polarization and enhance the mass transfer by disturbing the diffusive boundary layer [16,20,22,23], Vermaas et al. [24] through an experimental work showed that at low Re numbers (less than 100), which are typically used for RED, introducing non-conductive sub-corrugation is not that beneficial to reduce the ohmic losses and increase the power density [24]. They also showed that although the non-ohmic resistance (concentration boundary layer effects) decreases significantly when increasing the Reynolds number; the ohmic resistances are almost independent of the Re number at high Re numbers and dominates the power loss [24]. Pawlowski et al. performed an extensive literature review of the development and application of corrugated membranes in electro-membrane-based processes [25]. They reported the effect of corrugated membranes in the performance of reverse electrodialysis (RED), showing that electrodialysis (ED) is significantly influenced by the shape of the corrugation, Reynolds number, and ion concentrations. For high Reynolds numbers, corrugation creates eddies which lead to enhanced mass transfer, reduced deposition of foulants, and increased diffuse boundary layer thickness. In particular, they highlighted the role of conductive spacers in lowering the resistance of the RED stack, by eliminating spacer shadow effects [25]. They foresaw the rapid progress of the design and manufacturing of corrugated membranes due to advances in CFD simulations and 3D printing technology [25]. Gurreri et al. [26] used CFD modeling to study fluid flow behavior in a reverse electrodialysis stack, aiming to address the effect of the spacer material on the pressure losses along the channel, evaluating the choice of a fiber-structure porous medium, instead of the commonly adopted net spacers, and investigated the influences of the distributor and channel configurations on fluid dynamics in a RED system [26]. They documented that the total pressure loss in a RED stack is the sum of the pressure drop relevant to the feed distributor, the pressure drop inside the channel, and the pressure drop in the discharging collector [26]. Simulations revealed that the spacer geometry may not necessarily be the main factor controlling the overall pressure drop. In addition, the pressure drop induced by a porous medium made of small fibers is larger than that for a typical net spacer; therefore, they might not be suitable for RED [26]. Pawlowski et al. [27] showed, by CFD modeling, that chevron-corrugated membranes have the highest net produced power density among several investigated profiled membranes, due to increased membrane area, reduction of the concentration polarization, and the proper trade-off between momentum and mass transfer [27]. These results were validated also through experimental comparison [28]. Cerva et al. [29] presented a coupled study of one-dimensional CFD modeling with three-dimensional finite volume modeling for a flat channel, profiled membranes, and different spacer-filled corrugations in a RED stack. Then, they validated the overall model by comparison with experimental data measured in a laboratory [29]. Their results showed that the boundary layer potential drop is significantly lower than the ohmic losses. In addition, woven spacers had the smallest boundary layer potential loss, followed by Overlapped Crossed Filaments (OCF) profiled membranes and then the flat channel, thus indicating that woven spacers provide the most efficient and effective mixing among the considered systems [29]. The highest gross power density and the highest short-circuit current density were reported for OCF profiles, followed by the woven spacers and then the flat channel. However, the highest net power density per cell pair was provided by the flat channel, followed by OCF profiled membranes and then by the woven spacers [29]. Mehdizadeh et al. [30] experimentally studied several non-conductive spacers with different geometries and properties (e.g., different diameters, angles, distances, area fractions, and volume fractions) to understand the spacer shadow effect on the membrane and solution compartment resistances in RED. They reported a correlation between the spacer shadow effect on the membrane resistance and a combined parameter of spacer area fraction and spacer diameter [30]. The spacer shadow effect on the solution compartment resistance was also correlated with the spacer area and volume fraction. They observed that the spacer area fraction had a dominant effect only for less porous spacers [30]. Jalili et al. [31,32] used CFD modeling to examine the influence of flow velocities and spacer topology with respect to the transport of mass and momentum, as well as the flow channel resistivity of a RED unit cell. They reported that the resistivity of the dilute solution channel dominates over the resistivity of the concentrated solution channel and membranes in a RED unit cell [32]. Similar observations have also been reported by Ortiz-Martinez et al. [33]. The electrical potential of a RED unit cell was enhanced by reducing the flow velocity and introducing flow promoters in a dilute solution channel, due to reduced solution resistance [32]. Introducing spacers in a concentrated solution channel or increasing the flow velocity in a dilute solution channel increases the resistivity and has adverse effects on the electrical potential [32]. They also demonstrated that the mass transfer is higher for active membrane-integrated spacers, compared to inactive spacers, under similar flow velocity and spacer topology, due to increased active membrane area [31]. They also concluded that cylindrical membrane-integrated corrugation is an optimum spacer geometry at low flow velocities, while triangular membrane-integrated corrugation is a better geometry at high flow velocities [31]. Recently Dong et al. [34] performed a CFD study of mass and momentum transfer for several types of profiled membrane channels in RED. Their work showed that conductive wavy sub-corrugations improved the mass transfer and reduced the concentration polarization (i.e., non-ohmic losses) [34]. Furthermore, they showed that single-sided wave-profiled membranes had better performance, compared to single-sided pillar-profiled membranes; while single-sided profiled membranes had a smaller impact on the performance, compared to double-sided chevron-profiled membrane and woven spacer-filled channels [34].

Long et al. reported a numerical study matched with experimental data for optimizing channel geometry and flow rate of the concentrated and diluted solutions with non-conductive spacers, to obtain maximum net power output by RED. They reported that the optimal channel thickness and flow rate in the concentrated solution compartment in a RED stack are, respectively, much less than those of the dilute solution compartment [35]. In another work, they revealed that the optimal flow rates in the dilute and concentrated solution channels in an RED stack with varying flow rates along the flow direction to achieve maximum energy efficiency were lower than the optimal flow rates to obtain the maximum net power density. Therefore, an optimization study based on the Non-dominated Sorting Genetic Algorithm II (NSGA-II) was performed, in order to analyze the compromise between the net peak power density and the energy efficiency [36]. Their work showed that the net power density at maximum energy efficiency was less than the peak power density [36].

Several researchers have highlighted the potential use of waste heat in RED systems. Luo et al. [37] reported that by using ammonium bicarbonate as a working fluid in a thermally driven electrochemical generator, waste heat could be converted to electricity [37]. A maximum power density was obtained at an overall energy efficiency of 0.33 W m${}^{-2}$, by operating a RED system with a dilute concentration of 0.02 M [37]. Micari et al. [38] reported the conversion of waste heat into electricity by coupling RED with membrane distillation (MD), resulting in considerable system energy efficiency improvement. The construction and operation of the first lab-scale prototype unit of a thermolytic reverse electrodialysis heat engine (t-RED HE) for converting low-temperature waste heat into electricity have been reported by Giacalone et al. [39]. Ortiz-Imedio et al. [33] documented the strong dependence of the performance of RED on temperature. They reported that the membrane resistance increased when reducing the temperature, and that the perm-selectivity reduced when increasing the temperature [33]. Jalili et al. [31] showed that increasing the temperature enhanced the mass transfer of dilute and concentrated solutions, due to higher diffusivity and lower viscosity at increased temperature. In another work, they reported that the open-circuit potential increased with increasing temperature [2]. Contrary to the most of the literature, which has investigated salinity gradient energy at isothermal conditions, Long et al. [40] addressed the asymmetric temperature influence in dilute and concentrated solution channels on the performance of nanofluidic power systems, using numerical simulation by coupling the Poisson–Nernst–Planck equation and the Navier–Stokes equation, as well as the energy-conservation equation. They observed that when the temperature of the concentrated solution channel is lower than the temperature of the dilute solution channel, the ion-concentration polarization is suppressed, ion diffusion along the osmotic direction enhances, and perm-selectivity increases; thus, the membrane potential improves [40]. However when the temperature in the concentrated solution channel is higher than that of the dilute solution channel, the membrane potential reduces; although the diffusion current increases, due to the lower resistance [40]. In another work [41], they reported the influences of heat transfer and the membrane thermal conductivity in the performance of nanofluidic energy conversion systems. They reported that when the temperature of the concentrated solution channel is lower than the temperature of the dilute solution channel, a larger membrane thermal conductivity results, with reduced electrical power improvement; on the other hand, when the temperature of the concentrated solution channel is higher than the temperature of the dilute solution channel, the increased membrane thermal conductivity leads to enhanced power density [41].

Although several studies have reported the application of CFD modeling for investigating momentum and mass transfer in order to determine the trade-off between the pressure loss and mass transfer in an RED channel [16,18,19,22,23,27], there have been limited CFD studies of electrical potential in an RED channel [42,43]. To the best of our knowledge, there have been no parametric studies which assessed the relative effect of relevant parameters on the net power density for a RED cell. In particular, addressing the influence of temperature, as proposed by Jalili et al. [31], was not compared to the other parameters. The current work is an extension of the previously published works [31,32] by the current authors. We demonstrate that the electrical potential changes linearly with the height of the channel for a constant concentration profile, and that it follows a logarithmic trend with length of the channel height when the concentration profile varies linearly with the channel height [32]. Other interesting observations of this work [32] can be summarized as follows: First, the concentration gradient near the walls of the channel increase, due to reduced boundary layer thickness, with higher Re number. In fact, the concentration at the center of the channel is at its maximum for the concentrated solution channel and is at its minimum for the diluted solution channel [32]. Second, the pressure drop for the dilute solution channel is lower than that in the concentrated solution channel, given similar Re number and channel geometry [32]. This observation was also reported by Zhu et al. [21], when conducting several experiments. Third, the resistance of the dilute solution is more dominant, compared to the resistance of the concentrated solution channel, which can be seen as a limiting factor for the power density of a RED stack. Reducing the Re number (i.e., reducing the velocity at a constant temperature) or introducing corrugation in a dilute solution channel reduces the resistivity of the dilute solution channel by increasing the thickness of the boundary layer, which provides a thicker and more conductive region in the flow channel and results in improved mixing by the developing wakes downstream from the spacers [32]. An opposite trend was observed for the resistivity of the concentrated solution channel [32]. This observation was also supported by Long et al. [35].

This present work describes a numerical framework for simulation of the Navier–Stokes (NS) and Nernst–Planck (NP) system, based on the open source CFD platform OpenFOAM [44], with the aim of predicting the influence of flow velocity, temperature, and geometry on concentration, pressure drop, electrical potential drop, and net power density. Factorial design [45] is applied to address the relative effects of the parameters on the peak power density.

## 2. Theory and Governing Equations

The flow in the channel is considered to be two-dimensional, incompressible, steady-state, isothermal, and laminar. Physical properties such as density and viscosity are assumed to be constant. There is charge neutrality in the whole system, where only monovalent ions exist. The Navier–Stokes and Nernst–Plank equations [42,46,47] are presented by Equation (Equation 1) and Equation (Equation 2), respectively.
(1)$$\rho \vec{u}\cdot \nabla \vec{u}=-\nabla p+\mu \nabla^{2}\vec{u}.$$
(2)$$\nabla \cdot \left[D_{i}\nabla C_{i}-\vec{u}C_{i}+C_{i}\mu_{EP}\nabla \phi \right]=0,$$
for species *i*, where $C_{i}$ is the concentration ([mol/m${}^{3}$]), $D_{i}$ is the diffusivity ([m${}^{2}$/s]), $\vec{u}$ is the fluid velocity ([m/s]), and
(3)$$\mu_{EPi}=\frac{D_{i}z_{i}F}{RT}$$
is the electrophoretic mobility ([m${}^{2}$/Vs]), where $z_{i}$ is the valency, F=96,485.3 C/mol is the Faraday constant, R=8.314 J/K·mol is the universal gas constant, and *T* is the temperature (in Kelvin), while ϕ is the electrostatic potential ([V]).

Assuming two monovalent ionic species, denoted + and -, and using charge neutrality (i.e., $C_{+}=C_{-}=C$), Equation (Equation 2) can be written as [31,32]:(4)$$\begin{array}{ccc} & & (\vec{u}\cdot \nabla )C=\frac{2\cdot D_{+}\cdot D_{-}}{D_{+}+D_{-}}\nabla^{2}C\equiv D\nabla^{2}C, \end{array}$$
where D is the effective diffusivity for the salt and *C* is the concentration. The effective diffusivity is assumed to be a function of temperature, using the published data by Bastug and Kuyucak [48].

The electrical potential can be calculated from the conservation of electrical current density $\vec{j}$ [32],
(5)$$\nabla \cdot \vec{j}=0.$$

The electrical current density is obtained by a weighted sum of the charged species, resulting in
(6)$$\begin{array}{ccc} \vec{j} & = & F(D_{-}-D_{+})\nabla C-\frac{F^{2}C}{RT}(D_{+}+D_{-})\nabla \phi , \end{array}$$
where the advective flux cancels out, due to monovalent ions and charge neutrality. Combining Equations (Equation 5) and (Equation 6), we obtain the following relation [31,32]:(7)$$(\frac{D_{+}-D_{-}}{D_{+}+D_{-}})\nabla^{2}C=\frac{F}{RT}\nabla \cdot (C\nabla \phi ),$$
from which the electrostatic potential can be calculated, given a known concentration field in Equation (Equation 4). The proposed framework essentially consists of four one-way coupled equations—namely the incompressible Navier–Stokes Equation (Equation 1) which, together with continuity, determine the pressure and velocity fields; the concentration Equation (Equation 4), which essentially is an advection–diffusion equation with a known velocity; and, finally, the equation for the electrostatic potential (Equation 7), which is essentially reduced to a Poisson equation with a known source term. Given the domain and boundary conditions described in the following sections, the incompressible Navier–Stokes equations are solved by means of the simpleFoam solver in OpenFOAM, modified to account for concentration and potential following the steps described, for instance, in the openfoamwiki [49].

The trade-off between maximum produced electrical potential and the current density provides the peak power density. The peak power density, $P_{RED}^{peak}$ (W/m${}^{2}$), of a RED unit cell, the principal parameter of interest in the current work, can be expressed as follows: [5,11,14]:(8)$$P_{RED}^{peak}=\frac{1}{r_{unit\;cell}}\frac{E_{OCP}^{2}}{4},$$
where $r_{unit\;cell}$ and $E_{OCP}$ represent the area resistance of the unit cell and the open-circuit potential of the unit cell, correspondingly. The area resistance of the unit cell can be calculated by Equation (Equation 9) [5,14]:(9)$$r_{unitcell}=\left(r_{AEM}+r_{CEM}+r_{d}+r_{c}\right),$$
where $r_{AEM}$ and $r_{CEM}$ are the area resistances of the AEM and CEM, respectively, and $r_{c}$ and $r_{d}$ are the total area resistances for concentrated and dilute solution channels, respectively. The open-circuit potential depends upon the concentrations of dilute and concentrated channels as well as temperature, each of which are assumed fixed for a given setup in the current work. Assuming constant membrane properties, the only remaining variables are the area resistances of the channels. The total area resistance of the channels is calculated by dividing area-weighted average of electrical potential difference across the channel by the current density at the peak power density of RED unit cell, as shown by Equation (Equation 10) [14,32]:(10)$$r_{j}=\left|\frac{\Delta \tilde{\Phi }}{j}\right|,$$
where $r_{j}$ is the total area resistance (ohmic and non-ohmic) of the concentrated or dilute channels, *j* is the current density, and
(11)$$\Delta \tilde{\Phi }=\frac{1}{A_{AEM}}\int_{AEM}\phi \;dA-\frac{1}{A_{CEM}}\int_{CEM}\phi \;dA,$$
is the difference in area-weighted average of electrical potential Φ, calculated on the active membrane. The electrostatic potential across each channel, and thereby also the resistance, can be calculated based on the coupled Nernst–Planck and Navier–Stokes framework, presented in the theory and governing equations section. The formulation used in the current work accounts for both local values and gradients in concentration, and thus accounts for both ohmic and non-ohmic contributions. It should be noted that when dividing the potential drop by the imposed current, as in the above equation, non-ohmic contributions appear as an ohmic potential drop, although they are not of an ohmic nature [14].

When operating a RED system, the diluted and concentrated solutions are pumped through the compartments between the membranes, which inevitably leads to an energy loss. The required pump power density for each channel can be estimated by Equation (Equation 12) [14]:(12)$$P_{pump}=\Delta p\frac{Q}{A}=\Delta p\frac{H}{L}u,$$
where *A* is the membrane area, *Q* is the volumetric flow rate through the channel, *H* is the height of the channel, *L* is the length of the channel, *u* is the average velocity in the channel, and Δp is the pressure drop across the channel length which will be estimated through CFD modeling. To reduce ohmic energy losses in RED systems, the channel height should be as thin as possible; however, as this leads to increased pumping losses, there is a need to find an optimum value though. There are several factors affecting the optimal thickness of the inter-membrane distance, dictated by flow velocity, salinity and hydrodynamic pressure drops, but generally 50–300 μm is considered an optimum. This is for sterile particle free systems, but also fouling and other effects in nature can affects this further [2,50].

Given the energy consumption in the pump, the net peak power density can be calculated as:(13)$$P_{net}=P_{RED}^{peak}-P_{pump}^{total}.$$

In summary, the net peak power density can be calculated as follows:1.Coupled flow, concentration and potential fields are calculated through Equations (Equation 1)–(Equation 7).2.The potential difference across each channel is computed, allowing for determination the corresponding area resistances, as of Equations (Equation 11) and (Equation 10).3.Unit cell resistances and the peak power densities are calculated based on Equations (Equation 8) and (Equation 9).4.Pumping power is estimated using Equation (Equation 12), considering the flow velocities, and pressure drop from Equation (Equation 1).5.The net peak power density is finally computed as of Equation (Equation 13).

## 3. Simulation Setup

Flat and non-conductive spacer-filled channels with cylindrical or triangular corrugation are shown in Figure 2. Jalili et al. [32] reported that introducing flow promoters in a dilute solution channel improves the performance of a RED unit cell, while it has an adverse effect in the concentrated channel. Hence, the corrugated geometries were assumed for the dilute solution compartments, while the flat geometry was considered for concentrated solution compartments in this work.

The inlet concentrations for the channels were considered to be uniform and equal to 0.016 M (close to the salinity of brackish water) for the dilute solution channel and 0.484 M (close to the average salinity of seawater) for the concentrated solution channel.

### 3.1. Boundary Conditions

A constant molar flux, according to the following equation, was assumed in the current model [16,51]. This molar flux corresponds to a constant current density $\vec{j}$, from which the peak power density of the RED system can be obtained (see Equation (Equation 8)):(14)$$\vec{{i_{i}}^{m}}=\frac{{t_{i}}^{0}}{z_{i}F}\vec{j},$$
where $\vec{{i_{i}}^{m}}$ is the ionic flux of species *i* and ${t_{i}}^{0}$ is the transport number of species *i*. Assuming an ideal membrane from the perm-selectivity perspective and the transport properties for both cations and anions (of a monovalent binary electrolyte such as NaCl) in the solution for simplicity, we obtain [16,51]:(15)$$i_{IEM}=\pm \frac{0.5j}{F},$$
where the sign shows the incoming flux in the dilute channel or outgoing flux in the concentrated channel. Applying Fick’s first law of diffusion, as given in Equation (Equation 16), and substituting it into Equation (Equation 15), we obatin a constant concentration gradient, as shown in Equation (Equation 17).
(16)$$i_{IEM}=D\frac{\partial C}{\partial n},$$
where *n* is the normal direction to the wall, D is the effective diffusivity, and $i_{IEM}$ is representative of the ionic flux through the membrane. Equating Equations (Equation 15) and (Equation 16) provides us with the boundary condition for the concentration at the membranes: (17)$$\frac{\partial C}{\partial n}=\pm \frac{0.5j}{FD}.$$

The boundary condition for the electrical potential on the top membrane is
(18)$$\begin{array}{ccc} \nabla \phi & = & \frac{RT}{F^{2}C}\left[\frac{F(D_{-}-D_{+})\nabla C-\vec{j}}{\left(D_{+}+D_{-}\right)}\right]. \end{array}$$

Evidently, Equation (Equation 7) can be solved using the boundary conditions for the concentration and electrical potential, considering Equations (Equation 17) and (Equation 18).

The constant flux assumption is an approximation representing the features corresponding to an average concentration difference between the channels. Figure 3 shows the specified boundary conditions for different parts of the channel.

The value of the velocity at the inlet depends on the sought Reynolds number, and is given as a parabolic profile. The outlet is specified to atmospheric pressure. The membranes and spacers are set to no-slip conditions at the walls, and with zero gradient in pressure. In the case of the spacer-filled channel, the spacers are assumed to be non-ion conductive, with a corresponding zero flux boundary condition. The electric potential at the bottom wall of the channel is set to zero and the electrical potential on the top wall (active membrane) is calculated based on Equation (Equation 18).

### 3.2. Grid Dependence, Verification, and Validation

A grid dependence study was performed in our previous publication [32]. Local mesh refinement was used for different channel typologies, with extensive refinement near the wall of the channel and spacers, as shown in Figure 4.

Each of the simulations in the current work are based on the finest resolution identified in [32], with an average resolution of 1.13 and 0.25 μm in *x*- and *y*- directions, respectively, resulting in approximately 1 M (hexahedral) cells for the full domain. As shown in [32], this resolution introduces an error of less than 0.5%.

The flow behavior of the proposed framework was validated by comparison with the experimental measurements reported by Da Costa et al. [52] and Haaksman et al. [53]; as presented in [32]. The simulated pressure gradient with cylindrical corrugation was found to be somewhat lower than the pressure gradient for woven spacers, as reported by Gurreri et al. [26] at a given Re number; however, some discrepancies are expected, as Gurreri et al. considered the pressure drop in the collector and distributor of the RED stack, in addition to the main channel. The numerical results for the potential and concentration have been verified for a flat channel by comparison with the semi-analytical solution proposed by Lacey [43], for both dilute and concentrated channels (see, e.g., [32]), showing good agreement between the concentration profile and the corresponding electrical potential across the height of the dilute compartment and our numerical solution [32].

### 3.3. Numerical Settings and Configuration

All simulations presented in the following chapter were performed using the OpenFOAM version 4.1 software [44] on the IDUN cluster [54]. A summary of the numerical settings used in the current work are given in Table 1. The absolute residual for pressure, velocity, concentration, and electrical potential was set to $10^{-6}$, while the relative residual for the parameters was set to $10^{-4}$.

### 3.4. Factorial Design and Parametric Study

The influence of four quantitative parameters—inlet velocity, corrugation density, corrugation height, and temperature—on the resistivity and net peak power density were investigated. In addition to these four quantitative parameters, the effect of corrugation shape (cylindrical versus triangular) was considered to be a qualitative parameter. To determine the relative influence of each parameter on the power density, a parametric study was performed using a factorial design, as described by Montgomery [45]. The various factors and their corresponding levels are given in Table 2.

The corresponding values of the parameters for each geometry are given in Table 3. Notice that the Re numbers change, based on both the inlet velocity and the temperature, due to the change in viscosity. The pressure drop for Re numbers larger than 10 was so high that it resulted in a negative net peak power density in a unit cell and, therefore, the Re number in this study was limited to less than 10.

In the factorial design, each of the parameters (*m* parameters) were investigated at *n* levels, which gave us a set of n×m simulations, where the influence of each parameter, as well as their combined effect, could be determined. In this factorial design, the parameters were restricted to two levels, designated + and − (i.e., each parameter had a high and low level). Therefore, the results were restricted to a linear response for a given factor. There were 2${}^{4}$ designs for cylindrical and 2${}^{4}$ designs for triangular corrugation. Other fluid properties used for the current simulations are summarized in Table 4.

## 4. Results and Discussion

Figure 5 shows the concentration contour for a dilute solution channel with cylindrical spacers. The corrugation height (radius) was 0.1 mm and the distance between two successive corrugation centers was 0.6 mm. The figure also shows the results for two different average inlet velocities (u = 4.5 and 25.8 $\frac{\mathrm{mm}}{s}$) at two different temperatures (T = 25 ${}^{\circ }$C and 55 ${}^{\circ }$C).

Higher velocity and lower temperature resulted in less mixing of solutions and, therefore, lower average bulk concentration (shown by cold blue color in the concentration contour map in Figure 5b); thus, higher resistivities and lower power densities were expected. Enhanced mixing (higher average bulk concentration) was observed at lower velocity and higher temperature (shown by red and warmer blue colors of the concentration contour map in Figure 5c). The concentration profiles versus the height of the channel at the a−−b cross-section line is shown in Figure 5a, with a distance of X = 10.8 mm from the inlet of the dilute channel. Data sets from this line, for all geometries, are gathered and compared to each other in Figure 6. This was done for two different inlet average velocities (u = 4.5 and 25.8 $\frac{\mathrm{mm}}{s}$) and two different temperatures (T = 25 ${}^{\circ }$C and 55 ${}^{\circ }$C). Again, this figure confirms that at higher velocities and lower temperatures, the average concentration became lower. For all cases, the current density along the wall of the channel was considered constant (i.e., the current density for the peak power density), while the concentration along the walls was not constant, due to the imposed boundary conditions. Furthermore, the conductivity profiles for four cases versus the height of the channel (the a−−b cross-section in Figure 5a) are compared in Figure 7. The solution conductivities can be calculated using Equation (Equation 19), in which conductivity is a function of the concentration of the solution.
(19)$$\begin{array}{ccc} \sigma & = & \frac{F^{2}C}{RT}(D_{+}+D_{-}). \end{array}$$

The higher conductivity of the dilute channel agreed with the lower resistivity of the channel and, thus, a higher power density could be achieved. Figure 7 shows that the channel with lower flow velocity (u = 4.5 $\frac{\mathrm{mm}}{s}$) and higher temperature (T = 55 ${}^{\circ }$C) had enhanced mixing, with the highest calculated power density among these four cases.

### 4.1. Parametric Study

The performance of reverse electrodialysis is influenced by several parameters. Their single or combined impacts were investigated, using a parametric study and a factorial design. Results for the area resistance and power density are summarized for cylindrical corrugation in Table 5, and for triangular corrugation in Table 6.

The simulated net power densities found in this numerical study were comparable to the maximum power densities for RED reported by the authors in another publication [2], which were calculated by using conceptual analytical models with similar channel dimensions, as well as similar temperature and concentration ranges. In addition, the power densities obtained in this study at 25 ${}^{\circ }$C were close to the calculated power densities reported by Long et al. [40], at similar temperature and isothermal conditions, by applying numerical modeling for the investigation of nanofluidic salinity gradient energy harvesting [40]. The simulated net peak power densities were also in the range of the net power densities reported by Vermaas et al. [14], who calculated the theoretical RED net power density for different spacer-filled channels with channel thicknesses between 1–200 μm, and with residence time (defined as the length of the channel divided by the inlet flow velocity) between 0.5–200 s, in addition to changing the channel length and the resistivity of the AEM and the CEM. The residence time in the current work was within 0.2 to 2 s for the high- and low-level cases, respectively. As the resistivity of the channel decreased, the net power density for a RED unit cell increased for all system configurations. This observation was valid both for cylindrical and triangular corrugations, and was due to reduced lower-ohmic and non-ohmic losses. Increasing the temperature had a positive effect on the net peak power density, due to higher open-circuit potential, enhanced diffusivity, and improved mixing of concentrated and dilute solutions, as well as a lower pressure drop due to lower fluid viscosity at elevated temperature. Similar observations were reported experimentally by Luo et al. [37], Benneker et al. [57], and Daniilidis et al. [58]. Increasing the flow velocity had an adverse effect on the net power density, as a result of decreased mass transfer and increased pressure losses.

Vermaas et al. [55] also reported that the RED net power density was reduced for flows with Re numbers larger than 1 in channels with different thicknesses. The corrugation density and corrugation height had both positive and negative effects on the net peak power density. The corrugation height had an adverse effect on net power density, as pressure loss and consumed energy increase with higher corrugation height. This occurs even if the resistivity is lightly reduced, due to the increased corrugation height. In summary, one can conclude that the optimum parameters among the studied cases (i.e., for maximizing the net power density) was when the temperature was 55 ${}^{\circ }$C, the flow velocity was 4.5 $\frac{\mathrm{mm}}{s}$, the corrugation density was 20, and the corrugation height was 0.05 mm (for both the cylindrical and the triangular corrugation); see Table 5 and Table 6.

The triangular spacer corrugation configuration had slightly better performance, compared to the cylindrical one, which was in agreement with the previous studies reported by Ahmad et al. [20] and Jalili et al. [31]. The estimated effect of each factor is shown in Table 7 and Table 8. The tables reveal that temperature was the most dominant factor, followed by inlet velocity, corrugation density, and corrugation height, respectively.

It is worth mentioning that the simulated power densities in this study were larger than the experimentally measured power densities, such as those reported by Zhu et al. [59]. The current mathematical model was developed for incompressible, steady-state, isothermal, and laminar flow with only the presence of monovalent ions. Therefore, the results of the CFD model might not be representative when the flow regime is turbulent, the system is in unsteady state, or if multivalent ions exist. In addition, this CFD model is proposed for a unit cell; thus, it does not represent a full RED stack. The influences of anion and cation exchange membranes or water osmosis of the membranes are ignored in this study. Other sources of energy losses, such as pumping losses through the collector and distributor of the stack, are also neglected, as well as the practical issues relating to 3D flow distribution.

### 4.2. Concentration Polarization

The area resistances reported in Table 5 and Table 6 were calculated based on the electrical potential drop across the channel height for the whole channel, thereby accounting for both ohmic and non-ohmic contributions. By comparing the corresponding conductivities in Table 5 and Table 6 with the conductivities in Figure 7, in which only ohmic contributions are considered, we can obtain the non-ohmic contribution (i.e., the share of polarization in the system), as shown in Table 9. In fact, the resistivity calculated by Equation (Equation 10) is the area-weighted total resistivity which depends on the area-weighted electrical potential loss, and is obtained directly from solving Equations (Equation 1)–(Equation 7), provided the boundary conditions. Equation (Equation 19) provides the average conductivity of the channel solution based on the average concentration. The reverse of the average conductivity is the average ohmic resistivity. The difference between the total and the average ohmic resistivity, gives the average non-ohmic resistivity.

By comparing the resistivities in Table 9, three observations can be made: First, the share of non-ohmic losses (i.e., concentration polarization effects) was significantly lower than ohmic losses. Second, by increasing the flow velocity at a constant temperature or reducing the temperature at a constant inlet velocity, the ohmic losses increase. Third, increasing the flow velocity and temperature results in the reduction of the non-ohmic losses share of the total resistivity; that is, increasing the Re number (by enhancing the temperature or increasing the inlet flow velocity) will assist in reducing the concentration polarization effect in RED systems. These are consistent with the experimental observations reported by Vermaas et al. [55].

## 5. Conclusions

The effect of flow velocities, temperature, and spacer topology on the resistivity and net peak power density of a reverse electrodialysis (RED) unit cell were explained, based on CFD modeling which enabled the simulation of flow, pressure drop, concentration, electrical potential, and power density. Our parametric study revealed that while increasing the temperature and corrugation density had positive effects on the net produced power density, increasing the flow velocity and corrugation height had adverse effects. Among the studied parameters, temperature was the most dominating factor, followed by inlet velocity, corrugation density, and corrugation height, respectively. Increasing the temperature benefited the system performance by decreasing the non-ohmic resistance and the corresponding energy losses. Increasing the temperature also benefited the system performance by decreasing ohmic resistances. Moreover, elevating the temperature led to a system with a better performance increase than varying the flow velocity. The increase of temperature can be realized by use of low-grade waste heat, as discussed in [1] for instance.

## Figures and Tables

**Figure 1 membranes-10-00209-f001:**
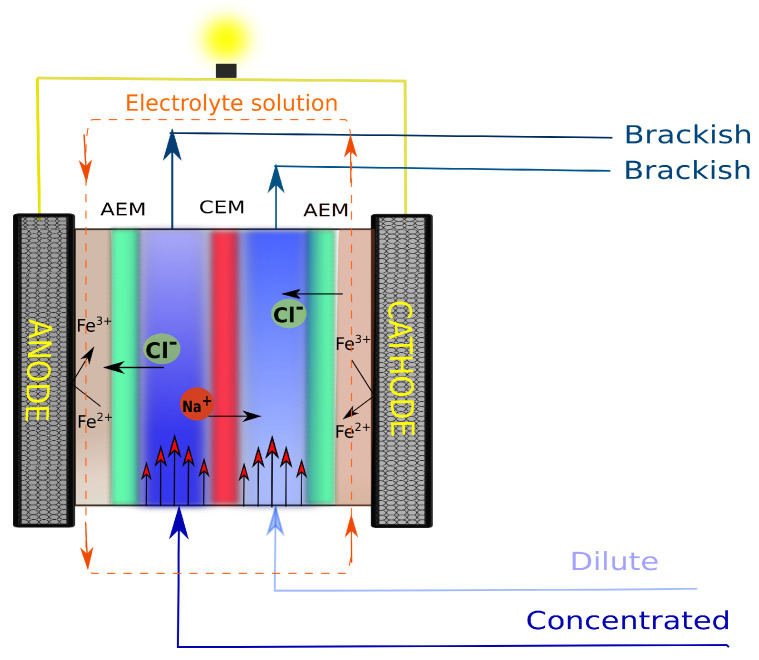
Schematic of a simple RED stack, containing (from the left) an anode, an anode electrolyte compartment, a unit cell, an additional membrane, a cathode electrolyte compartment, and a cathode.

**Figure 2 membranes-10-00209-f002:**
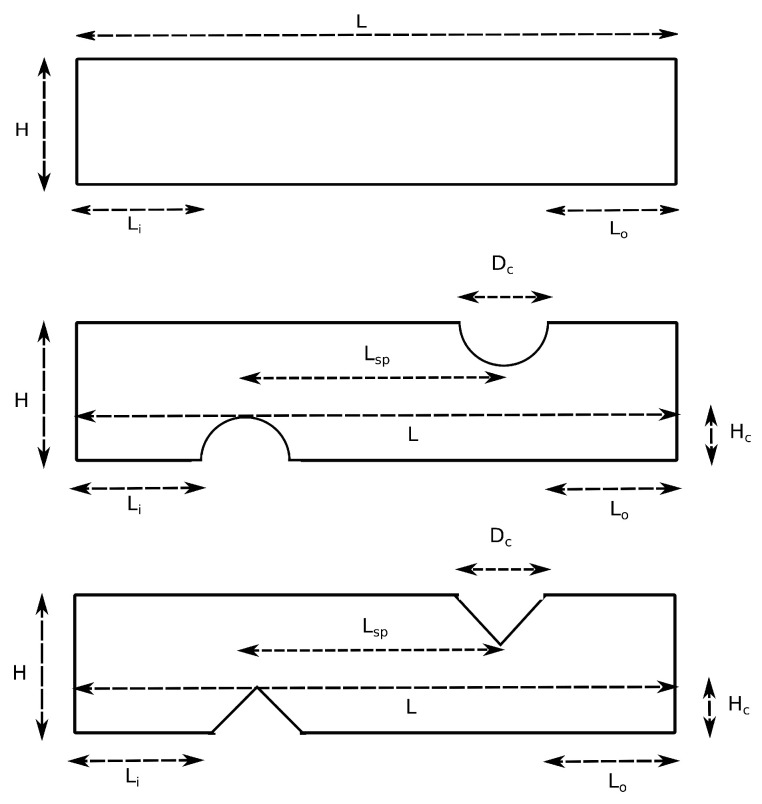
Schematic presentation for sections of the geometry of flat, cylindrical, and triangular corrugated channels with characteristic length scales.

**Figure 3 membranes-10-00209-f003:**
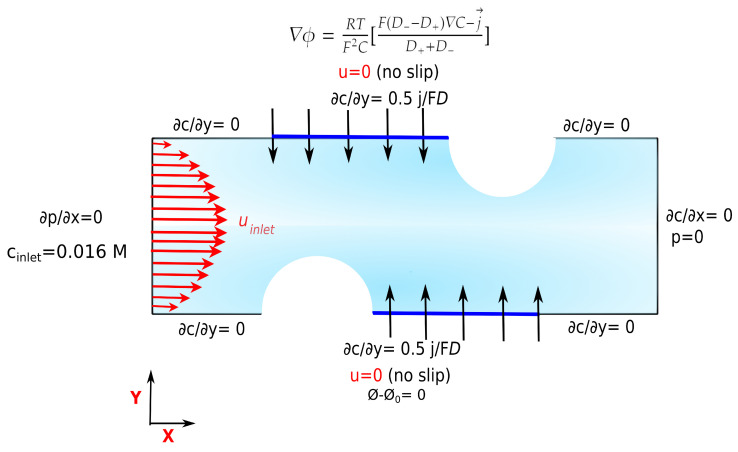
The boundary conditions for a section of dilute, non-conductive cylindrical spacer-filled channel. The blue line shows the active membrane section and the arrows show the diffusion direction from the top and bottom wall toward the dilute bulk. The geometry is repeated to build the full length of the compartment.

**Figure 4 membranes-10-00209-f004:**
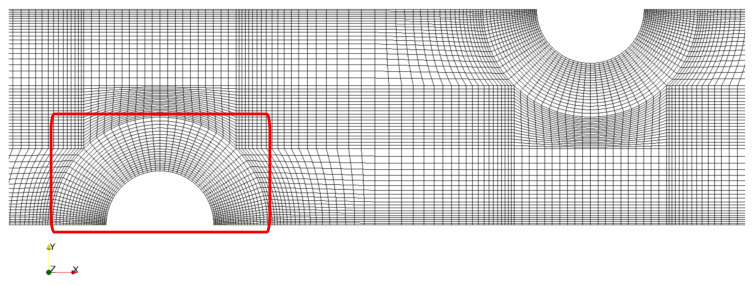
Local grid refinement near the walls of the cylindrical corrugated spacer-filled channel. The coarsest mesh in the local grid refinement process was depicted due to better visibility. The region around the corrugation which goes under local refinement process is confined in a red square.

**Figure 5 membranes-10-00209-f005:**
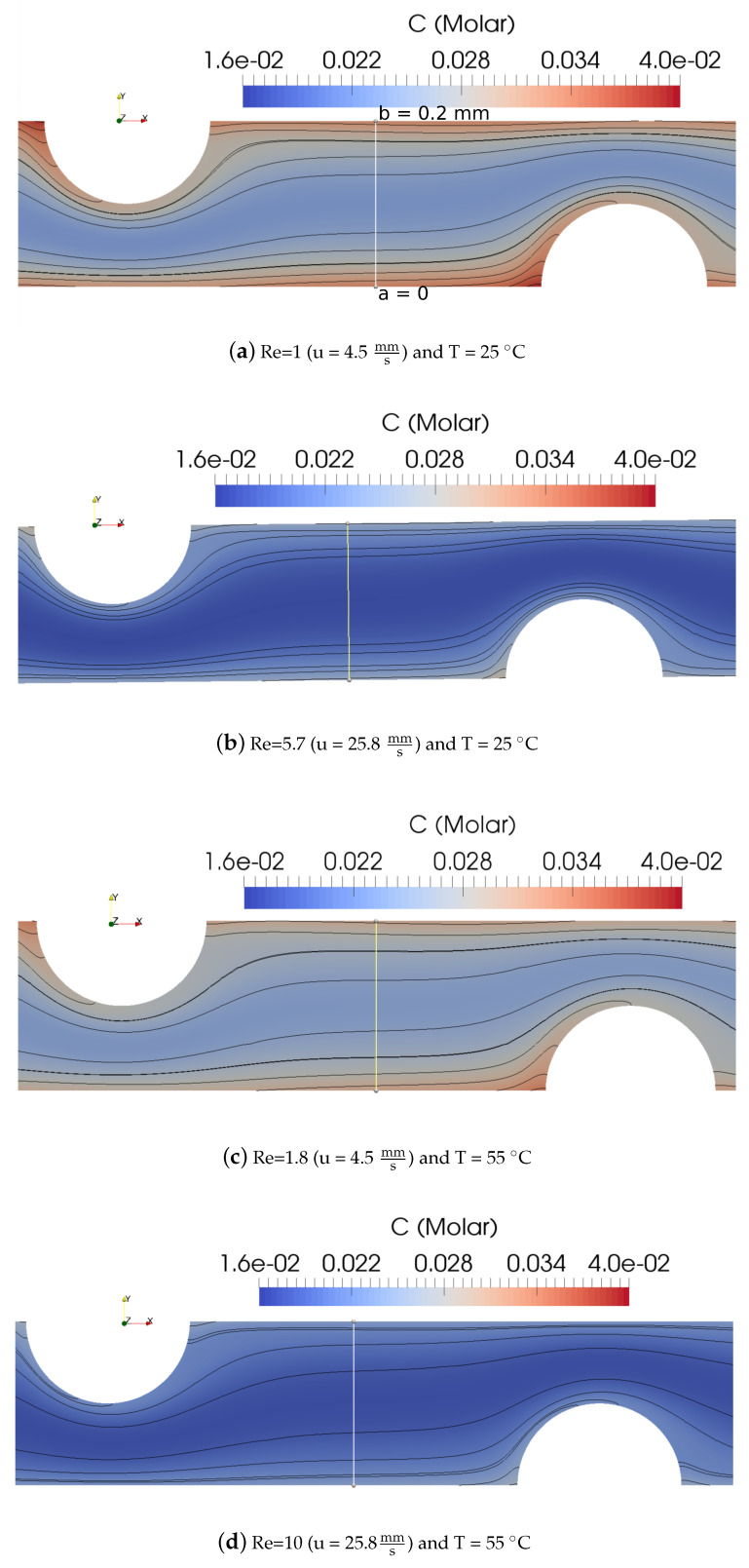
Concentration contour maps for dilute solution in a cylindrical corrugated channel at different Re numbers and temperatures.

**Figure 6 membranes-10-00209-f006:**
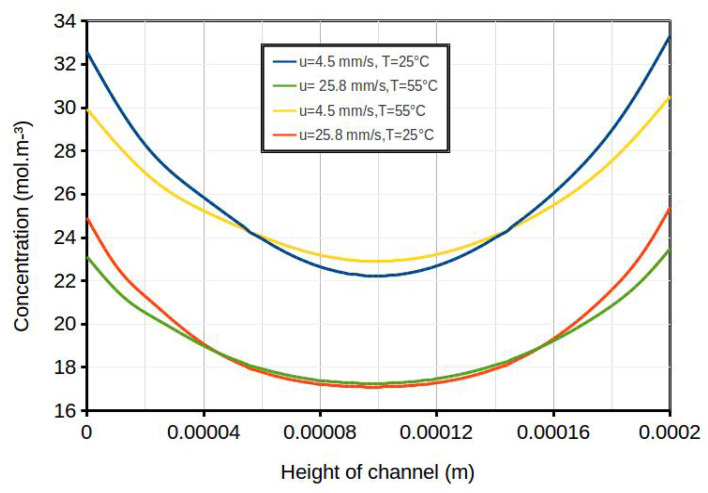
Concentration profiles versus the height of the channel (the a−−b cross-section in Figure 5a) at X = 0.0108 m from the inlet of the dilute channel at two different inlet average velocities (u = 4.5 and 25.8 $\frac{\mathrm{mm}}{s}$) and T = 25 ${}^{\circ }$C and T = 55 ${}^{\circ }$C, resulting in four different Re numbers: Re = 1, 1.8, 5.7, and 10.

**Figure 7 membranes-10-00209-f007:**
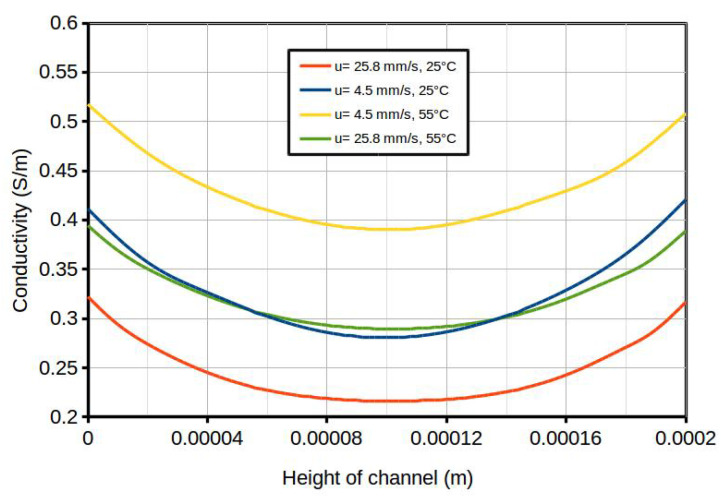
Conductivity versus the height of the channel (the a−−b cross-section in Figure 5a) at X = 0.0108 m from the inlet of the dilute channel at two different inlet average velocities (u = 4.5 and 25.8 $\frac{\mathrm{mm}}{s}$) and T = 25 ${}^{\circ }$C and T = 55 ${}^{\circ }$C, resulting in four different Re numbers: Re = 1, 1.8, 5.7, and 10.

**Table 1 membranes-10-00209-t001:** Discretization schemes specified for the case studies.

Term	Scheme
Time	steadyState
Gradient	Gauss, linear
Divergence	Bounded, Gauss, linearUpwind
Laplacian	Gauss, linear, corrected

**Table 2 membranes-10-00209-t002:** Factors and levels used for the $2^{4}$ design for cylindrical and triangular corrugated channels.

Factor	Name	High Level (+)	Low Level (−)
Inlet velocity	A	0.0258 m/s	0.0045 m/s
Temperature	B	55 ${}^{\circ }$C	25 ${}^{\circ }$C
Corrugation Density and $L_{sp}$	C	20 and 600 μm	16 and 800 μm
Corrugation Height	D	100 μm	50 μm

**Table 3 membranes-10-00209-t003:** Characteristic parameters of the studied geometries in factorial design and the input parameters. (The values of the current densities are dependent of the available area of the membranes for different topologies).

Parameter	Symbol		Value
Corrugation diameter	$D_{c}$		0.1 or 0.2 (mm)
Length of the channel	*L*		12.6 (mm)
Height of the channel	*H*		0.2 (mm)
Number of corrugations	*N*		16 or 20 (dimensionless)
Height of the corrugation	$H_{c}$		0.05 or 0.1 (mm)
Length of inlet and outlet section	$L_{i}$, $L_{o}$		0.25 and 0.85 (mm)
Distance of two successive corrugations center	$L_{sp}$		0.6 or 0.8 (mm)
Resistance of AEM and CEM	$r_{\mathrm{AEM}}$, $r_{\mathrm{CEM}}$		1.0 × $10^{-4}\;\Omega$ m${}^{2}$ [55]
Current densities	j		66, 68, 70 and 75 Am${}^{-2}$

**Table 4 membranes-10-00209-t004:** Transport properties of the fluid at temperatures considered, reported diffusivities by Bastug and Kuyucak [48], and viscosities by Tseng et al. [56].

T (K)	$D_{-}$ ($\frac{m^{2}}{s}$)	$D_{+}$ ($\frac{m^{2}}{s}$)	ν ($\frac{m^{2}}{s}$)
298	$2.03\times 10^{-9}$	$1.33\times 10^{-9}$	$9.05\times 10^{-7}$
328	$2.80\times 10^{-9}$	$2.10\times 10^{-9}$	$5.16\times 10^{-7}$

**Table 5 membranes-10-00209-t005:** Summary of factors, area resistance of the dilute solution compartment, and net peak power densities of the unit cell in the 2D model of a **cylindrical** corrugated channel: A, velocity; B, temperature; C, corrugation density; D, corrugation height. Case 1 is the base case.

	Factor				Response	
Name	A	B	C	D	Area Resistance (Ω·cm${}^{2}$)	Net Peak Power Density (W/m${}^{2}$)
**Case 1**	−	−	−	−	7.15	6.18
**Case 2**	+	−	−	−	8.56	5.43
**Case 3**	−	+	−	−	5.47	8.86
**Case 4**	+	+	−	−	6.51	7.96
**Case 5**	−	−	+	−	7.02	6.26
**Case 6**	+	−	+	−	8.40	5.50
**Case 7**	−	+	+	−	5.37	8.95
**Case 8**	+	+	+	−	6.39	8.05
**Case 9**	−	−	−	+	7.91	5.80
**Case 10**	+	−	−	+	9.45	5.02
**Case 11**	−	+	−	+	6.05	8.35
**Case 12**	+	+	−	+	7.19	7.48
**Case 13**	−	−	+	+	7.88	5.82
**Case 14**	+	−	+	+	9.41	5.03
**Case 15**	−	+	+	+	6.04	8.36
**Case 16**	+	+	+	+	7.16	7.46

**Table 6 membranes-10-00209-t006:** Summary of factors, area resistance of the dilute solution compartment, and net peak power densities of the unit cell in the 2D model of a **triangular** corrugated channel: A, velocity; B, temperature; C, corrugation density; D, corrugation height. Case 1 is the base case.

	Factor				Response	
Name	A	B	C	D	Area Resistance (Ω·cm${}^{2}$)	Net Peak Power Density (W/m${}^{2}$)
**Case 1**	−	−	−	−	7.08	6.23
**Case 2**	+	−	−	−	8.47	5.47
**Case 3**	−	+	−	−	5.41	8.91
**Case 4**	+	+	−	−	6.44	8.01
**Case 5**	−	−	+	−	6.92	6.27
**Case 6**	+	−	+	−	8.28	5.45
**Case 7**	−	+	+	−	5.29	9.02
**Case 8**	+	+	+	−	6.29	8.12
**Case 9**	−	−	−	+	7.52	5.99
**Case 10**	+	−	−	+	8.99	5.22
**Case 11**	−	+	−	+	5.76	8.60
**Case 12**	+	+	−	+	6.84	7.72
**Case 13**	−	−	+	+	7.37	6.07
**Case 14**	+	−	+	+	8.80	5.29
**Case 15**	−	+	+	+	5.64	8.70
**Case 16**	+	+	+	+	6.69	7.80

**Table 7 membranes-10-00209-t007:** Sign and percent contribution of area resistance and power density for each of the factors in the **cylindrical** corrugated channel shown in Table 5.

**Factor**	**A**	**B**	**AB**	**C**	**AC**	**BC**	**ABC**	**D**
Sign Area resistance	+	−	−	−	−	+	+	+
%	26.6	62.5	< 1	< 1	< 1	< 1	< 1	9.90
**Factor**	**AD**	**BD**	**ABD**	**CD**	**ACD**	**BCD**	**ABCD**	
Sign Area resistance	+	−	−	+	+	−	+	
%	< 1	< 1	< 1	< 1	< 1	< 1	< 1	
**Factor**	**A**	**B**	**AB**	**C**	**AC**	**BC**	**ABC**	**D**
Sign Power density	−	+	−	+	−	+	−	−
%	9.29	87.5	< 1	< 1	< 1	< 1	< 1	3.10
**Factor**	**AD**	**BD**	**ABD**	**CD**	**ACD**	**BCD**	**ABCD**	
Sign Power density	−	−	+	−	−	−	−	
%	< 1	< 1	< 1	< 1	< 1	< 1	< 1	

**Table 8 membranes-10-00209-t008:** Sign and percent contribution of area resistance and power density for each of the factors in the **triangular** corrugated channel shown in Table 6.

**Factor**	**A**	**B**	**AB**	**C**	**AC**	**BC**	**ABC**	**D**
Sign Area resistance	+	−	−	−	−	+	+	+
%	28.3	67	< 1	< 1	< 1	< 1	< 1	3.46
**Factor**	**AD**	**BD**	**ABD**	**CD**	**ACD**	**BCD**	**ABCD**	
Sign Area resistance	+	−	−	+	+	+	+	
%	< 1	< 1	< 1	< 1	< 1	< 1	< 1	
**Factor**	**A**	**B**	**AB**	**C**	**AC**	**BC**	**ABC**	**D**
Sign Power density	−	+	−	+	−	+	+	−
%	9	90	< 1	< 1	< 1	< 1	< 1	< 1
**Factor**	**AD**	**BD**	**ABD**	**CD**	**ACD**	**BCD**	**ABCD**	
Sign Power density	+	−	+	+	+	−	−	
%	< 1	< 1	< 1	< 1	< 1	< 1	< 1	

**Table 9 membranes-10-00209-t009:** The contribution of ohmic and non-ohmic resistance for Cases 13, 14, 15, and 16 of Table 5 (i.e., with cylindrical corrugation).

Resistivity (Ω·m)	Case 13	Case 14	Case 15	Case 16
**Total**	3.94	4.71	3.02	3.58
**Ohmic**	3.08	4.09	2.36	3.13
**Non-ohmic**	0.86	0.61	0.66	0.45

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
