# Peer review of "Computational Fluid Dynamics Modeling of the Resistivity and Power Density in Reverse Electrodialysis: A Parametric Study"

_membranes, 2020, doi:10.3390/membranes10090209_

Round 1

Reviewer 1 Report

The topic sounds exacting. Combining Nernst-Planck and Navier-Stokes equations in an open-source code such as OpenFoam could be of interest to readers. Open-source platforms lead to faster and easier spread of scientific knowledge and progress. However, the code written by the authors to combine NP and NS equations is actually not provided, thus the main interest behind this manuscript is not made available, as the presented findings were already previously presented, either based on experimental or mechanistic modelling evidence. However, even if the code would be provided, the manuscript needs first a very deep English revision and the manuscript requires a better care. There are many typos in the text, references are not by numeric order, there are references which are miss referenced or switched and there are many sentences which needs revision.

Reviewer 2 Report

The authors explored the effects of Reynolds number, membrane topology, and temperature on the RED performance using numerical modeling via CFD software OpenFOAM. This manuscript shows some novel results and the authors are suggested to make revisions to address a few comments listed below:

  1. Figure 3 shows concentration contours for the dilute solution at different Re numbers under two temperature levels 25oC and 55oC, respectively. The authors are suggested to make a comparison under the same Re number and two levels of temperature 25oC and 55oC, which would be better to explore the effect of temperature on the concentration distribution. In addition, the contour lines are too small and nearly invisible in some places. The unit of the legend should be listed and the unit should be consistent in either mm or m.
  2. In the Eq. (6), the flux includes diffusion flux and electromigration flux. However, the authors tend to solve a coupling of Navier-Stokes and Nernst-Plank equations. Please explain why the advection flux term is ignored. Also, please add the Navier-Stokes equations used in the manuscript.
  3. The authors do not show clearly the value of diffusivity coefficients D+ and D- for the governing equations from Eq. (4) to Eq. (7). Those values seem to be constant in their setting. However, the diffusivity coefficient should vary as a function of temperature. The effect of temperature on RED performance has been recently reported by V-P Mai and R-J Yang (Please see Applied Energy. 274, 115294 and RSC Advances, 10, 18624- 18631).
  4. The authors are suggested to depict numerical settings in the manuscript (not shown in their previous article published in ESC Transaction) including the mesh type, the number of mesh, and mesh independent solution, et al.

Reviewer 3 Report

This study deals with a parametric study on the resistivity and power density in reverse electrodialysis. The topic is interesting, and the manuscript is well-written. It can be acceptable for publication after following revisions.

  1. The numerical model should be validated.
  2. The grid-independence should be included.
  3. In the schematic diagram, the boundary conditions should be added to offer a better reader ship.
  4. The cation and anion concentration profiles can be included to offer a better illustration of the ion distribution in the channels.
  5. Some literatures regarding the reverse electrodialysis in a single nanopore in CFD may could broaden the readership, such as “ Ionic thermal up-diffusion in nanofluidic salinity gradient energy harvesting” and “Effects of heat transfer and the membrane thermal conductivity on the thermally nanofluidic salinity gradient energy conversion”.
  6. Some recent studies regarding the optimization of RED systems such as “multi-objective optimization for the flow rates “Reverse electrodialysis: Modelling and performance analysis based on multi-objective optimization (2018)” and optimization for the channel thickness “ Performance analysis of reverse electrodialysis stacks: Channel geometry and flow rate optimization (2018)” could be included. The impacts of flow rates are equal to those of the Reynolds numbers, which are deeply investigated in present manuscript. Above recent literatures may help better understanding the impacts of the Reynolds number on the system performance.

Round 2

Reviewer 1 Report

There are still points which needs attention:

  • Line 22-23: Ref. 2 is not representative regarding the use of RED “in nature”. Papers reporting the use of real waters (natural saline streams), in real conditions, should be cited instead to prove the applicability of using RED to harvest SGE.
  • Ref. 14 is about the benefits of electrodialysis for desalination. I cannot see how the authors cite ref. 14 to claim RED is better than PRO.
  • The neutrality condition in Fig. 1 is not respected.
  • Line 50 – this statement should be supported by a reference, and it should be explained at which conditions optimal thickness is between 50-300 um, since fouling/clogging can shift these values.
  • Why concentration changes in the boundary layer, and concentration changes in the bulk solutions are considered as non-ohmic resistances? Non-ohmic resistance means that ohm law is not valid. However, eqs.1-4 are all of them for the case in which ohm law is valid, so these two things are not compatible.
  • Ref. 27 and 28 are the same. Ref. 27 should be Pawlowski et al. Int. J. Mol. Sci. 2019, 20, 165. I do not know what should be the correct ref. 28, since the paper which is cited as ref. 28 is not about profiled membranes. Moreover, that sentence: “In addition, introducing corrugation leads to higher pressure loss, due to more significant hydraulic friction through the channel; thus, lower net power densities are produced from the process (28).” is not correct. There are cases in which introduction of corrugations increase/decrease pressure drop, increase/decrease net power density, and it all depends on the shape, dimensions, distribution of corrugations and spacers to whom the comparison is made (as mentioned by the authors in the sentences written in lines 74-94).
  • The conclusions about the effect of sub-corrugations are different in refs. 36 and 26. The authors should discuss such apparent contradictions. The lack of such discussion results also from +- chronological organization of introduction. It would be clearer if there would be part of the introduction focus on profiled membranes and other part focused on spacers. Now, it is all mixtured, which sometimes made difficult to understand if the authors are referring to profiled membranes or to case in which spacers were used. For example, from the text it is not possible to understand what was studied by Long et al. (line 119).
  • Line 233: How the local mesh refinement was done? Perhaps it is an optical illusion, but the finest refinement appears to be in the middle of the channel, which is strange, as the main transfer phenomena occurs near membrane surface, and, therefore, the highest refinement should be applied there.
  • Table IV: simpleFoam is a steady state solver, so how it is possible to table iv be a summary of the numerical settings differing from the default simpleFoam values if it is reported in table iv that steadyState scheme was used for time ? At least that condition is not different from the default. The manuscript improved a lot, but it still contains a lot of contradictions, misreferred papers, and it would be much more beneficial to readers could see a case example folder and code of the developed solver, instead of reading about it, especially because it is possible as the OpenFoam is an open-source tool.

Round 3

Reviewer 1 Report

Lines 22-23 and further – How ref. 2 demonstrate that “Among the different methods for harvesting salinity gradient energy, reverse electrodialysis (RED) could be more interesting, as reported by Jalili et al. (2).” ? In ref. 2 the reported costs of electric energy generation, for the same membrane price, are the lowest for PRO. And it is known that the price of membranes for RED is higher than those for PRO, therefore, looking at Fig. 24a, the best choice appears to be PRO, at least at 25ºC. Only if the temperature increases, RED performance might overcome PRO. However, the authors are mentioning in lines 33-34, that the considered system, and higher generated power density by RED are for river and seawater system. How 60ºC can be achieved for river-seawater system ? Are the cost of heating included in the analyses? Finally, the work performed by Tedesco et al. was performed with brines, and not for river-seawater system. And, comparing with other studies in the literature, brackish-brine systems, due to higher concentrations differences, outperform river-seawater systems, therefore the authors should review very carefully what was written in lines 22-43.

Figure 1 – Electroneutrality conditions is not respected.

Eq.2 – How rd and rc are calculated ? Are they constant along channel ? Or are they calculated locally ? Since it is a CFD study, the values of concentration are calculated at each volume. Thus, why that so called “non-ohmic resistance” due concentration polarisation? If the local values of concentrations are known, it is possible to calculate local values of resistances. Which are all ohmic, so why concentration polarization is considered as non-ohmic ? (But only in name, because for calculations it is assumed that ohm law is valid ….). How the results obtained by CFD simulations are used in Eqs. 1-6 ? How eqs. 1-6 are employed ?

Introduction of corrugations instead of spacers might lead to decrease of pressure drop. That comment was made in function of what authors wrote previously, and not in comparison to pressure drop in a flat channel. I am not aware of any RED stack working with flat/empty channels, without spacers or corrugations. If the authors are comparing empty channels with channels with spacers or corrugations it is not clear in the manuscript.

Round 4

Reviewer 1 Report

As authors noticed in their replay, the number of the required revisions has been high. This is the 4th time that I will give my opinion/comments. As stated in the first revision, it is an exciting topic, but the manuscript itself contained a lot of faults, which step by step, although sometimes it was/is needed to ask twice, are being corrected, but very slowly, and by doing the minimum of improvements. The core of the manuscript, CFD studies and the modelling approach are finally explained +- satisfactory, thus I’m not going to propose the rejection of the paper. But I would like to ask to the authors to do a deep revision of the work, of the references which are cited, rearrangement of the manuscript structure, because I really hope to could accept the paper to be published after the 5th version.

The submitted manuscript is about CFD modelling of RED stack. Thus, the introduction should be about that topic. The authors instead try to explain first why RED is better than other technologies, which is unnecessary. And in their explanations mix different operation modes under which RED can be performed, and clearly are unable to have a critical view on papers which they are citing, since as it is shown in their answers, they just limited to report sentences written in those papers. The problem is that the progress in the RED field was extraordinary fast during last decade. In 2007, only river-seawater was so far tested at that time, and for such a system, in 2007, it was concluded that RED is better than PRO. Since than different pairs of streams, configurations, arrangements, corrugations, spacers, etc., were tested. All of this, if RED or PRO are better, is irrelevant and not needed in the introduction of the paper, but if the authors wish to write a paragraph about it, please have a critical analysis, based on the most recent papers.

Moreover, I felt extremely concerned with the answer given by the authors about work of Tedesco et al. If the authors want to show the benefits of RED for river-sea water system, it is incorrect to mention in the middle of such a paragraph a a work in which the system under study was brackish-brine waters and refer to such a work that “Tedesco et al. reported a full-scale RED pilot plant for power generation with natural streams in a real condition (14).” If somebody does not know his work, it is very easy to conclude, erroneously, that “natural streams in real condition” is a river-seawater system. Such sentence can be very easily clarified by saying which streams were used instead of saying “natural streams”. If the authors want to “prove” RED is better than PRO because a pilot plant was operated, then well actually you cannot say that because there was as well a pilot PRO installation somewhere in Norway. I am not sure, but both these plants, of RED and PRO are no longer working. There is a Dutch RED plant which, I think, is still operational. However, as I was saying, nothing of this is relevant for what was done by the authors, which were CFD studies. But, if the authors wish to claim that RED is better than PRO, you are free to do it, but the used arguments should be well supported, and a critical analysis should be done instead of citing 13-year old sentences which are the most convenient for the given hypothesis. This manuscript is about using CFD tools in RED. It does not matter if RED is, or is not, the best technology to harvest SGE. But if such a claim is made, it should be fully supported. 

Eqs. 2 and 3: To clarify: Are rd and rc calculated by eq. 3 ? Is rj rd or rc if calculated for dilute and concentrated channel, respectively? My doubt results from the utilization of the word “total” (the total area resistance of the channels) when eqs. 3 is introduced, because it is mentioned that “rc and rd are the area resistances” (without the word “total”). Please check my last comment before startting talking in rbl and rdc resistances.

Line 69: For the clarity, the manuscript would be easier to be followed, if eqs. 1-6 would be presented after CFD in section 2. It requires some additional work from the authors to reorganise the structure. I sincerely believe, and it is a friendly comment, that manuscript would be clearer if CFD approach would be presented first, as eqs. 1-6 use the data computed by CFD simulations. Eqs. 1-6 are very well known equations and do not need to be presented in the introduction, but is very important to be clear how CFD results were employed in eqs. 1-6. Please check also my next and the penultimate comments.

Line 73: “Pumping losses are estimated based on the flow velocities” and pressure drop in the channel computed in CFD simulations …Am I correct ? How that pressure drop was computed ?

Lines 95-97: “In addition, introducing corrugation leads to higher pressure loss, compared to the pressure loss for flat channel, due to more significant hydraulic friction through the channel, given other similar conditions (31).” I completely do not understand reason of this sentence. Obviously, the pressure drop in an empty channel will be lower than in any other channel geometry. Ref. 31 do not study the presence of corrugations. And, finally, what are “other similar conditions” ? It is very unclear. Please remind that introduction of corrugations can increase pressure drop, or it can decrease pressure drop, in comparison to any channel with spacers! It all depends of the dimensions, shape, distribution of corrugations. It also depends of the spacer to which the comparison is done! There is no universal sentence that can be said about this topic! Please remove or rewrite.

Section 3.4 should be before section 3.3.

Eqs.1-6 should be presented after section 3.4 and before section 3.3. The explanation given in the answer “The resistivity calculated by equation 3 is the area weighted total resistivity which depends on area weighted electrical potential loss, which is obtained directly from solving equations 7-13, provided the boundary conditions in the manuscript. The values reported by Equation (19) are average ohmic resistivity which depends on the average concentration. The difference between these two, is the average non-ohmic resistivity.” should be somehow included in the manuscript as it is important for the reader.

“Equation 3 in our manuscript relates the potential loss to the total resistivity (i.e. sum of the ohmic and non ohmic resistivity) and is based on the equations 3 and 4 in the work reported by Vermaas et al. (Ref. 17). However in their work, they calculated the ohmic and non ohmic resistsivities through analytical estimations, while we applied CFD modeling to obtain the electrical potential loss. “ – Exactly, you did CFD, David only had macroscopic data and the resistance measured on an entire stack. So, he did not have information about variation of the concentration in the channel, neither perpendicular, neither parallel, to the membrane. Thus, he included a mathematical way of including these contributions by including rdC and rBL. RdC was calculated. But rBL is the difference between experimental results and all other resistance which were calculated, and from its empirical correlations were derived. So, despite being associated to BL, it contains everything else which might exist and was not accounted, including the errors (experimental or even of the modelling approach since Vermass used correction factor to account for the effect of the spacers). You do not need to do it because by CFD it is possible to compute local concentrations (the information which Vermass did not have). But it is still unclear for me why to call them as non-ohmic contributions, despite Vermass called them that way, and afterwards still applying Ohm law. You do not need to replay to this point, unless you have some interesting explanation.
